# Bacterial and Fungal Endophytes of Grapevine Cultivars Growing in Primorsky Krai of Russia

Olga A. Aleynova [1,*], Nikolay N. Nityagovsky [1], Alexey A. Ananev [1], Andrey R. Suprun [1], Zlata V. Ogneva [1], Alina A. Dneprovskaya [1,2], Alina A. Beresh [1,2], Alexandra S. Dubrovina [1], Pavel A. Chebukin [3] and Konstantin V. Kiselev [1]

[1] Laboratory of Biotechnology, Federal Scientific Center of the East Asia Terrestrial Biodiversity, Far Eastern Branch of the Russian Academy of Sciences, Vladivostok 690022, Russia
[2] The School of Natural Sciences, Far Eastern Federal University, Vladivostok 690090, Russia
[3] Federal Scientific Center of Agricultural Biotechnology of the Far East Named after A.K. Chaika, Ussuriisk 692539, Russia
[*] Correspondence: aleynova@biosoil.ru; Tel.: +7-4232-310718; Fax: +7-4232-310193

**Abstract:** In this study, the biodiversity of endophytic bacteria of cultivated grape varieties from the vineyards of Primorsky Krai, Russia, was analyzed for the first time. Far Eastern grape varieties with a high level of stress resistance are a unique object of research as they are cultivated in cold and humid climates with a short summer season. Grapevine endophytic microorganisms are known as promising agents for the biological control of grapevine diseases and agricultural pests. Using genomic approaches, we analyzed the biodiversity of the endophytic bacteria and fungi in the most common grape varieties of Primorsky Krai, Russia: *Vitis vinifera* × *Vitis amurensis* cv. Adele (hybrid No. 82-41 F$^3$), *Vitis riparia* × *V. vinifera* cv. Mukuzani (pedigree unknown), two cultivars *Vitis labrusca* × *V. riparia* cv. Alfa, and *Vitis* Elmer Swenson 2-7-13 cv. Prairie Star for the first time. The main representatives of the endophytic microorganisms included 16 classes of bacteria and 21 classes of fungi. The endophytic bacterial community was dominated by Gammaproteobacteria (31–59%), followed by Alphaproteobacteria (9–34%) and, to a lesser extent, by the classes Bacteroidia (9–22%) and Actinobacteria (6–19%). The dominant fungal class was Dothideomycetes (43–77%) in all samples analyzed, with the exception of the grapevine cv. Mukuzani from Makarevich, where Malasseziomycetes was the dominant fungal class. In the samples cv. Alfa and cv. Praire Star, the dominant classes were Tremellomycetes and Microbotriomycetes. A comparative analysis of the endophytic communities of the cultivated grape varieties and the wild grape *V. amurensis* Rupr. was also carried out. We found that 18–20% of the variance between the endophytic communities accounted for the differences between the cultivated and wild grapevines, while the factors "plant location" and "individual plants" accounted for 50–56% and 3–10% of the variance, respectively. The results of this study can be used to develop new means of biocontrol in vineyards to protect plants from abiotic stresses and pathogens.

**Keywords:** grapes; metagenome; bacteria; fungi; endophytes; Far East of Russia; *16S*; *ITS1*; *Vitis amurensis*; NGS





## 1. Introduction

Grapevine is among the oldest cultivated plant species, being one of the most highly consumed fruit crops in the world [1]. The grapevine belongs to the Vitaceae family, consisting of approximately 14 genera and 900 species [2]. The genus *Vitis* represents more than 7.5 million ha of cultivated surfaces in the world with 27 million tons of wine produced each year [3]. The grapevines consist of multiple compartments like roots, leaves, and fruits, each providing distinct habitats for microorganisms. The microbiota found within different plant compartments is influenced by the grape genotype, development, and local

environment. However, the extent to which the microbiota in one compartment, such as the roots, shapes the microbiota in other parts like the leaves or fruits is not well known. Several factors, including the plant's geographic location, genotype, and biotic/abiotic stresses, influence the composition and diversity of plant microbiota. Previous research has revealed that specific bacterial and fungal communities are retained within grapevines compartments, irrespective of grafting or the rootstock genotype used [4]. This suggests that the environmental conditions experienced by microorganisms in different parts of the plant play a crucial role.

The most serious problems faced in the cultivation of grapes are abiotic and biotic stresses, which lead to a decrease in grape yield and fruit quality [5–7]. Biocontrol is gaining popularity in viticulture as a means to reduce the use of chemical pesticides, which have negative effects on the environment and human safety [8,9]. The grapevine endophytic microbiome, consisting of bacteria and fungi that inhabit plant tissues without causing harm, holds great potential as a source of biocontrol agents. These endophytes can be found in both natural and managed ecosystems [10]. Endophytes have the ability to produce beneficial metabolites like phosphorous, iron, and nitrogen from the environment, as well as growth-regulating phytohormones that help plants withstand abiotic stresses [11]. Additionally, endophytic microorganisms occupy a similar ecological niche as many plant pathogens, making them suitable candidates for biocontrol agents. Exploiting these endophytes as biocontrol agents could be an effective strategy to reduce pesticide usage in vineyards [3].

For example, the grapevine endophytic bacteria *Bacillus velezensis*, *Pseudomonas chlororaphis*, and *Serratia plymuthica* constitute potential biocontrol agents against grapevine trunk diseases [5]. Furthermore, the endophytic strain *Bacillus velezensis* KOF112, isolated from the Japanese wine grape *Vitis* sp. cv. Koshu, inhibits the growth of several aggressive plant pathogens such as *Botrytis cinerea* (gray mold), *Colletotrichum gloeosporioides* (causing anthracnose), and *Phytophthora infestans* (some of the most aggressive and widespread plant pathogens) [12]. In addition to bacteria, endophytic fungi also play a role in promoting plant growth. *Phialocephala fortinii*, for instance, has been found to increase phosphorus accumulation and biomass in *Vitis macrocarpon* [13]. These endophytes have the potential to enhance the overall health and productivity of grape plants. Moreover, endophytic bacteria like *Pseudomonas fluorescens* RG11 have been found to promote plant growth and increase the levels of melatonin, an important plant hormone, in different grape cultivars [14]. This hormone is known to aid in plant growth and can be particularly useful in helping plants withstand salt stress conditions [15]. By utilizing melatonin-producing endophytes, it may be possible to develop strategies that support plant growth under challenging environmental conditions. Another notable example is *Burkholderia phytofirmans* PsJN, a rhizobacterium that promotes plant growth and induces resistance to gray mold in grapevines. Additionally, it triggers physiological changes that enhance the grapevine's tolerance to low non-freezing temperatures. This bacterium shows promise as a biological agent for disease control and improving grapevine resilience to abiotic stresses [16]. Thus, endophytic microorganisms of grapes have a number of favorable properties for plant growth, also they have the potential to be utilized as biological agents in grape cultivation, contributing to sustainable agriculture practices and integrated plant production methods.

*Vitis vinifera*, a commonly cultivated grape species, is not able to survive harsh winters in regions with extremely low temperatures like northern China and Primorsky Krai in Russia. However, certain wild *Vitis* species such as *Vitis amurensis* and *Vitis riparia* have shown great tolerance to freezing conditions [7]. Despite this, the specific mechanisms responsible for grapevine cold tolerance are still largely unknown. Perhaps, in addition to the physiological and molecular features of grapes, endophytic microorganisms may be a key element in resistance to adverse environmental factors. The microbiome processes essential for vine growth and wine production exhibit distinct patterns linked to the location of the vineyard [17,18]. Currently, in the viticulture of Primorsky Krai, bred or imported grape varieties are used, capable of yielding a good harvest in conditions of a short summer

period, high relative humidity, and cold winter. It is known that the grapevines have been extensively manipulated through centuries of cultivation and breeding, resulting in highly selected germplasms [19]. Thus, information about the microbiome of Far Eastern grape cultivars is an interesting object of research.

Understanding the composition of the microbiome of grapevine varieties grown in the vineyards of Primorsky Krai of Russia will make it possible to correct the biodiversity and species ratio of endophytic microorganisms, which may contribute to improving grapevine growth and production characteristics. Therefore, the aim of this work was to study the biodiversity of endophytic bacteria and fungi in the most common grape cultivars growing in the vineyards of the Primorsky Krai of Russia using next-generation sequencing (NGS).

## 2. Materials and Methods

### 2.1. Samples Collection and Pre-Treatment

Healthy and normally developed grapevine tissues were collected from two different vineyards in Primorsky Krai of Russia in July 2022. In particular, from each site, Makarevich (longitude 43.718600 and latitude 132.11040) and PRIM ORGANICA (longitude 44.218073 and latitude 132.475822), were collected young shoots with three leaves of two 10-year-old grapevines of two cultivars *V. vinifera* × *V. amurensis* cv. Adele (hybrid No. 82-41 F$^3$), *Vitis riparia* × *V. vinifera* cv. Mukuzani (pedigree unknown) and two cultivars *Vitis labrusca* × *V. riparia* cv. Alfa (https://www.vivc.de/index.php?r=passport%2Fview&id=346, accessed on 18 October 2023), and *Vitis* Elmer Swenson 2-7-13 cv. Prairie Star (https://www.vivc.de/index.php?r=passport%2Fview&id=23087, accessed on 18 October 2023), respectively. In order to control weeds in the two vineyards, Makarevich and PRIM ORGANICA, a combination of manual weeding and mechanical processing of the aisles was employed. Additionally, approximately 5 kg of humus derived from cattle manure was used as a mineral fertilizer every two years during spring time. To prevent fungal diseases, particularly mildew, two fungicides were utilized in the PRIM ORGANICA vineyard: Thanos (containing famoxadone and cymoxanil) from DuPont, Wilmington, DE, USA, and Orden (containing copper chloride and cymoxanil) from August, Russia. These treatments were applied 3–4 times throughout the season, alternating between the two preparations. For defense against spider mites, the PRIM ORGANICA vineyard used the inorganic contact fungicide and acaricide Tiovit Jet (containing colloidal sulfur) from August, Russia, which exhibits high gas phase activity. In the Makarevich vineyard, three different treatments were employed to defend against spider mites, anthracnose, and mildew: Abiga Peak (containing copper chloroxide) from Abiga Peak, Russia, Ridomil Gold (containing ethylene bisdithiocarbamate and mefenoxam) from Syngenta, Switzerland, and Kurzat R (containing copper chloroxide) at a specified concentration. Insect protection was provided by using two insecticides: Fufanon (containing malathion) from BuFF Fufanon, Germany, and Actara (containing thiamethoxam) from Syngenta, China. These insecticides were employed to safeguard the vineyards against harmful insects. Overall, these vineyard processing techniques and the use of various fertilizers, fungicides, and insecticides help maintain the health and productivity of the vineyards by controlling weeds, preventing fungal diseases, and protecting against harmful insects.

The plant material was collected at 11–12 a.m. on low-cloud days without precipitation, and the air temperature was 18–20 °C. Each plant material specimen was delivered to the laboratory within 3 h. Four biological replicates (two stems and two leaves) of each of the grapevine cultivars were collected and analyzed using a cultivation-independent approach (NGS).

To prepare the grapevine tissues for further analysis, each cultivar's tissues (0.5 g) were cleaned using a specific procedure. First, the tissues were washed with soap under running water. Then, they were sequentially washed under sterile conditions using 75% ethanol for 2 min, 10% hydrogen peroxide for 1 min, and finally rinsed five times with sterile water. This process aimed to effectively sterilize the surface of the tissues [20,21]. To verify the success of this sterilization method, a sample of the last wash water (100 μL)

was incubated on R2A and potato dextrose agar (PDA) plates. The purpose was to ensure that there was no bacterial or fungal colony growth on these plates, indicating the absence of any contamination from the outside (*in vitro* control of epiphytic microorganisms). This sterilization procedure is crucial for maintaining the integrity and purity of the grapevine tissues, preventing any unwanted microorganisms from interfering with subsequent analysis or experiments.

### 2.2. DNA Extraction and Illumina MiSeq Sequencing

The DNA for NGS was isolated using the method as described earlier [22]. The quality and quantity of the DNA was assessed using the NanoPhotometer P300 (IMPLEN, Munich, Germany).

The DNA samples were sent to Sintol (Moscow, Russia) for Illumina high-throughput sequencing. The libraries were prepared for sequencing following the protocol described in "*16S* Meta-genomic Sequencing Library Preparation" (Part # 15,044,223 Rev. B; Illumina (San Diego, CA, USA)). The bacterial *16S* rRNA V4 region (515Fmod-806R) was amplified from all samples using plant primers modified for *Vitis* sp. 515F (5′GGTAATACGKAGGKKGCD-AGC) and 806R (5′RTGGACTACCAGGGTATCTAA) [20]. The *ITS1-ITS2* rDNA region of the fungi was amplified from all samples using primers ITS1f (5′CTTGGTCATTTAGAGGA-AGTAA) and ITS2 (5′GCTGCGTTCTTCATCGATGC) [21]. Once the amplicons had been obtained, the libraries were purified and mixed in an equimolar ratio using the Sequal-Prep™ Normalization Plate Kit (ThermoFisher, Waltham, MA, USA, Cat # A10510-01). The resulting library pools were quality-checked using the Fragment Analyzer, and quantitative analysis was performed using qPCR. The library pool was sequenced on Illumina MiSeq (2 × 250 paired end) using the MiSeq Reagent Kit v2 (500 cycles). The FASTQ files were obtained using bcl2fastq conversion software v2.17.1.14 (Illumina). The phage PhiX library was used to control the sequencing parameters. The majority of the reads that belonged to the phage DNA were removed during the demultiplexing process.

Bacterial and fungal sequences of the endophytes were deposited into NCBI under the accession number PRJNA998468 and in the database of the Biotechnology laboratory at the Federal Scientific Center of the East Asia Terrestrial Biodiversity, Far Eastern Branch of the Russian Academy of Sciences, Russia (https://biosoil.ru/downloads/biotech/Vitis%20metagenom/, accessed on 18 October 2023).

### 2.3. Bioinformatics and Biostatistics

The samples used in the bioinformatic analysis are presented in Supplementary Materials Tables S1 and S2. Custom scripts based on the R and Bash languages were used to process the data obtained (https://github.com/niknit96/Aleynova_et.al.2023.10, accessed on 18 October 2023). The QIIME 2 [23] and DADA2 [24] programs were used to pre-process the raw data. Paired-end reads were merged and sorted to remove primers, remaining PhiX reads, and chimeric sequences. Taxonomic identification of sequences was performed using the QIIME 2 Scikit-learn algorithm using the SILVA 138 pre-trained classifier for *16S* sequences (99% OTUs from V4 region of sequences) [25] and the UNITE pre-trained classifier for *ITS* sequences (99% OTUs from *ITS1f*/*ITS2* region of sequences) [26].

The qiime2R [27], phyloseq [28], ggdendro [29], RColorBrewer [30], circlize [31], and tidyverse [32] libraries were used in the pre-filtering and data preparation. Amplicon sequence variants were merged into genus-level taxonomic ranks. Mitochondria, chloro-plast, Viridiplantae, Metazoa, Rhizaria, Protista, Alveolata, and unidentified sequences were deleted from the obtained data. Taxa at the genus level were filtered on the basis of relative abundance > 0.1% for a plant. We merged the filtered genera into a group called "other" in the taxonomy bar plots at the class level. Also, "other" genus taxa were removed from the UpSet diagrams.

The tidygraph [33], ggraph [34], Netcommi [35], and SpiecEasi [36] R libraries were used to analyze the positive and negative associations between bacteria and fungi in the *Vitis* endophytic community. Bacterial and fungal genera that did not occur more

than 3 times in at least 60% of the samples were removed to reduce sparsity and ensure reliable results. The genus-level taxa were normalized, transformed, and converted into an adjacency matrix based on covariance by the package SpiecEasi using Meinshausen and Bühlmann neighborhood selection to estimate the conditional dependence of each pair of genera. This approach is suitable for compositional amplicon data and, unlike correlation-based methods, prevents most spurious, indirect relationships from being included in the networks [36]. The adjacency matrix was then converted into a Tidygraph object with help the Netcommi package and visualized as a circular network using the ggraph library. The centrality of nodes was measured using Kleinberg's hub centrality scores. The top 10% of taxa of genus level based on the values of Kleinberg's hub centrality score were qualified as hubs.

The Shannon alpha diversity and Bray–Curtis beta diversity data were obtained using the Vegan package [37]. The Bray–Curtis dissimilarity data were transformed into an even sampling depth and converted into non-metric multidimensional scaling (NMDS). To analyze the alpha diversity data between groups, a pairwise Wilcoxon rank-sum test with false discovery rate correction was performed. The PERMANOVA test with 999 permutations was used to statistically validate the beta diversity data. For the graphical representation of the results, the ggplot2 [32] and ComplexHeatmap [38] R libraries were used.

## 3. Results

### 3.1. Illumina Next-Generation Sequencing Results

Using Illumina technology, all *16S* rRNA V4 regions (*16S*) and *ITS1-ITS2* rDNA regions (*ITS1*) of the DNA samples were successfully sequenced, and libraries were constructed. NGS produced a total of 3,285,397 *16S* and 743,375 *ITS1* paired reads, respectively. After paired-end alignments, quality filtering, and the deletion of singletons and chimeric and non-bacteria or non-fungal sequences, a total of 1,990,927 bacterial *16S* and 262,933 fungal *ITS1* sequences were generated from 4 plant samples (4 samples from each plant). On average, 124,433 and 16,433 sequences, per grapevine sample, were obtained from the *16S* and *ITS1* regions (Supplementary Materials Tables S1 and S2).

### 3.2. The Biodiversity of Bacterial Endophytes from Grapevine Cultivars in Primorsky Krai of Russia

After the bioinformatic quality control procedures, a total of 1,990,927 *16S* sequences in 16 samples and 172 taxa of genus level with a relative representation above 0.1% were identified, which belong to 18 taxa of class level. The endophytic bacterial community composition was mostly matched by class Gammaproteobacteria (28–59%) followed by class Alphaproteobacteria (8–35%) and, to a lesser extent, by classes Bacteroidia (6–22%) and Actinobacteria (5–21%) (Figure 1a). The biodiversity of the endophytic bacteria in grapevines cv. PrairieStar and cv. Alfa from the vineyard PRIM ORGANICA was most similar in the percentage of bacterial classes, as were the cv. Adele and cv. Mukuzani from the vineyard Makarevich (Figure 1a). The leaf and stem tissues of the grapevine were similar in terms of the bacterial biodiversity within the plant based on UPGMA clustering.

We also compared the biodiversity of the endophytic bacteria of grape varieties with the bacterial endophytic community of wild grapes *V. amurensis*; the data were obtained in previous research [39]. According to the analysis, in all samples of cultivated grapes, except for the cv. Adele, there was a similar percentage of classes of endophytic bacteria. In the grapevine cv. Adele, the percentage of the main Gammaproteobacteria class increased up to 59% and the percentage of the Alphaproteobacteria reduction class decreased up to 8–10% (Figure 1a).

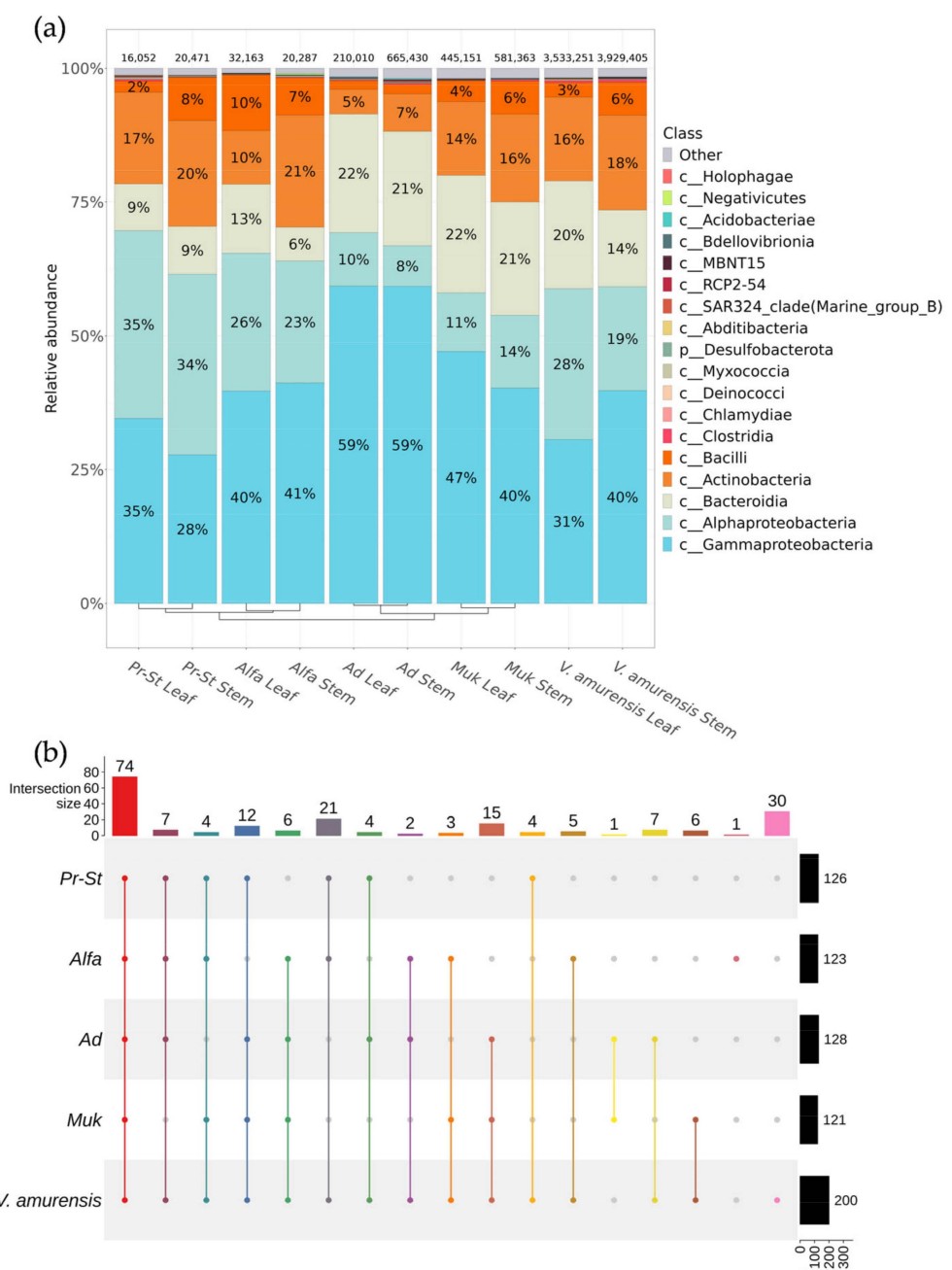

**Figure 1.** Comparative analysis of the bacterial endophytic community composition of grapevine cultivars and wild grape *Vitis amurensis* growing in Primorsky Krai of Russia according to genomic approach (NGS). The composition of endophytic bacteria of grapevine variates Pr-St—*Vitis* Elmer Swenson 2-7-13 cv. Prairie Star from commercial vineyard PRIM ORGANICA; Alfa—*Vitis labrusca* × *Vitis riparia* cv. Alfa from PRIM ORGANICA; Ad—*Vitis vinifera* × *V. amurensis* cv. Adele from commercial vineyard Makarevich; Muk—*V. riparia* × *V. vinifera* cv. Mukuzani; *V. amurensis*—the sum of data for all *V. amurensis* grapevines obtained early [39]: (**a**) class-level taxonomical bar plots for the bacteria endophytic community depends on the variates of grapevines, and the sum of data for all *V. amurensis* grapevines obtained early [39]; (**b**) genus-level UpSet diagrams depicting overlapping taxa of NGS in different cultivars of grapevines. For each biocompartment, taxa were filtered based on relative abundance > 0.1%. Taxa with relative frequencies < 0.1% were removed from the UpSet plot. Number of sequences are shown above taxonomical bar plots. For clustering in bar plots, we used unweighted pair group method with arithmetic mean (UPGMA).

According to the analysis, the endophytic bacterial community in the studied grape cultivars was represented by 121–128 genera of bacteria, while the community of endophytic bacteria in wild grape *V. amurensis* was represented by 200 genera (Figure 1b). Among them, 74 taxa were common for all grapevines (Figure 1b and Supplementary Materials Table S3). At the same time, 30 genera of endophytic bacteria were unique for the wild *V. amurensis* grape. The representation of the endophytic genera in the grape varieties collected at the Makarevich vineyard was similar to the grape samples from PRIM ORGANICA. Each sample of the studied grape variety contained 4–7 overlapping genera with the wild *V. amurensis* grape (Figure 1b). The grapevine cv. Alpha from PRIM ORGANICA had one unique genus, *Eikenella*, of endophytic bacteria (Figure 1b and Supplementary Materials Table S3).

The most common taxa for all analyzed grapevines were the taxa Comamonadaceae, *Sphingomonas* and *Methylobacterium-Methylorubrum* (Figure 2). For the grapes cv. Adel and cv. Mukuzani from the Makarevich vineyard, a large percentage were the bacterial genera *Aquabacterium* and *Spirosoma*, as well as *Asinibacterium* and *Ralstonia* being represented in these grape varieties and in the wild grape *V. amurensis* and absent in the grapevines from PRIM ORGANICA (Figure 2). The representation of endophytic bacteria *Abiotrophia* and *Neisseria* was 4 and 9% in the grape cv. Alpha from PRIM ORGANICA, while the representation of these bacterial genera in other varieties and the wild grape *V. amurensis* was less than 0.1% (Figure 2).

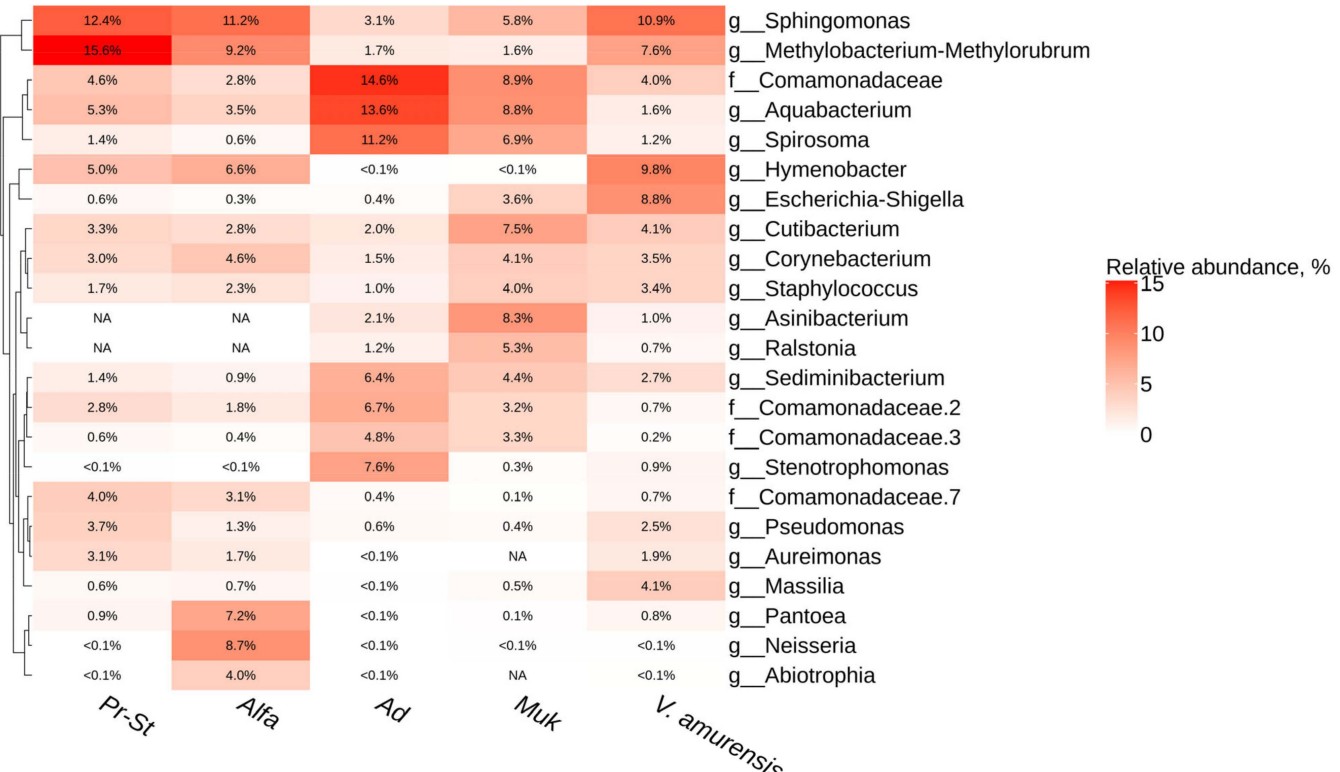

**Figure 2.** Genus-level relative endophytic bacteria abundance heat map of significant taxa according to next-generation sequencing (NGS) in grapevines cultivars and wild grape *Vitis amurensis*. Pr-St—*Vitis* Elmer Swenson 2-7-13 cv. Prairie Star from commercial vineyard PRIM ORGANICA; Alfa—*Vitis labrusca* × *Vitis riparia* cv. Alfa from PRIM ORGANICA; Ad—*Vitis vinifera* × *V. amurensis* cv. Adele from commercial vineyard Makarevich; Muk—*V. riparia* × *V. vinifera* cv. Mukuzani; *V. amurensis*—the sum of data for all *V. amurensis* grapevines obtained early [39]. The top 10 most abundant taxa from each grapevine are displayed. White squares (NA) represent the absence of taxa.

### 3.3. The Fungal and Fungi-like Endophytic Microorganisms from Grapevine Varietes in Primorsky Krai of Russia

A total of 262,933 *ITS1* reads were used for the phyla description for fungi and fungi-like endophytes of the vine varieties. According to metagenomic analysis of the *ITS1* sequences, 146 taxa of genus level with a relative representation above 0.1% were represented in the community of fungi and fungi-like endophytes in different grapevines. These genera belonged to 24 taxa of class level in the analyzed grapevines (Figure 3a). The dominant class of fungi was Dothideomycetes (22–79%) in all analyzed samples except the grapevine cv. Mukuzani and the stem tissue from cv. Adel from Makarevich. In the grapes of the Mukuzani variety, the class Malasseziomycetes was dominant. In addition, the presence of Oomycetes (6–23%) and Agaricomycetes (4–14%) was found in the samples collected at the Makarevich vineyard. In the samples of grapes from PRIM ORGANICA, the predominating classes were Tremellomycetes and Microbotryomycetes (Figure 3a). The leaf and stem tissues of cv. Prairie Star and cv. Alfa were similar in terms of their intra-plant fungal biodiversity based on UPGMA clustering, while the stem tissue in cv. Adel was similar to the leaf and stem tissues in cv. Mukuzani.

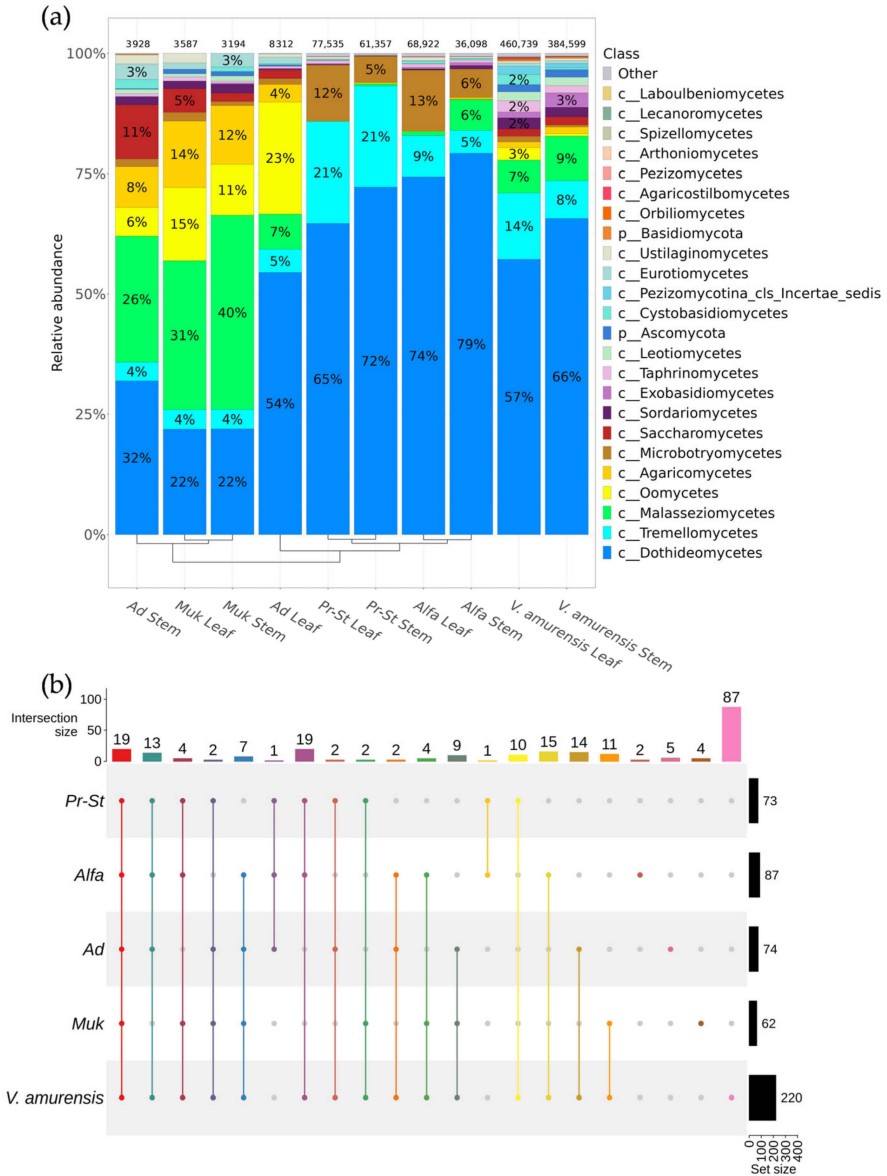

**Figure 3.** Comparative analysis of the fungal and fungi-like endophytic community grapevine cultivars and wild grape *Vitis amurensis* growing in Primorsky Krai of Russia according to genomic

approach (NGS). The composition of endophytic fungi and fungi-like microorganisms of grapevine variates Pr-St—*Vitis* Elmer Swenson 2-7-13 cv. Prairie Star from commercial vineyard PRIM ORGANICA; Alfa—*Vitis labrusca* × *Vitis riparia* cv. Alfa from PRIM OR-GANICA; Ad—*Vitis vinifera* × *V. amurensis* cv. Adele from commercial vineyard Makarevich; Muk—*V. riparia* × *V. vinifera* cv. Mukuzani; *V.amurensis*—the sum of data for all *V. amurensis* grapevines obtained early [39]: (**a**) class-level taxonomical bar plots for the fungal and fungi-like endophytic community depend on the variates of grapevines, and the sum of data for all *V. amurensis* grapevines obtained early [39]; (**b**) genus-level UpSet diagrams depicting overlapping taxa of NGS in different cultivars of grapevines. For each biocompartment, taxa were filtered based on relative abundance > 0.1%. Taxa with relative frequencies < 0.1% were removed from the UpSet plot. Number of sequences are shown above taxonomical bar plots. For clustering in the bar plots, we used UPGMA.

The representation of the genera of fungi and fungi-like microorganisms varied from 62 to 87 in the cultivated grape varieties, while the representation of genera in the wild grape *V. amurensis* was 220 genera (Figure 3b and Supplementary Materials Table S4). Among them, 19 taxa were found in all grapevines. The largest number of taxa of genus level were present in the grapevine cv. Alfa from PRIM ORGANICA (87 genera). The 87 fungi and fungi-like genera were unique for wild grape *V. amurensis*, 2 for grapevine cv. Alfa from PRIM ORGANICA, and 4–5 for each cultivar of grape from the Makarevich vineyard (Figure 3b and Supplementary Materials Table S4).

The most dominant taxa for the grape cv. Adel and cv. Mukuzani from the Makarevich vineyard were *Malassezia*, *Aureobasidium*, and *Plasmopara* (Figure 4). For the grapevines from PRIM ORGANICA, the predominant genera were *Alternaria*, *Aureobasidium*, and *Cladosporium* (Figure 4). The genera *Kalmanozyma* and *Pseudopithomyces* were representative for the grapevine cv. PrairiStar, cv. Alfa from PRIM ORGANICA, and wild grape *V. amurensis*, and absent in the grapevine from Makarevich (Figure 4).

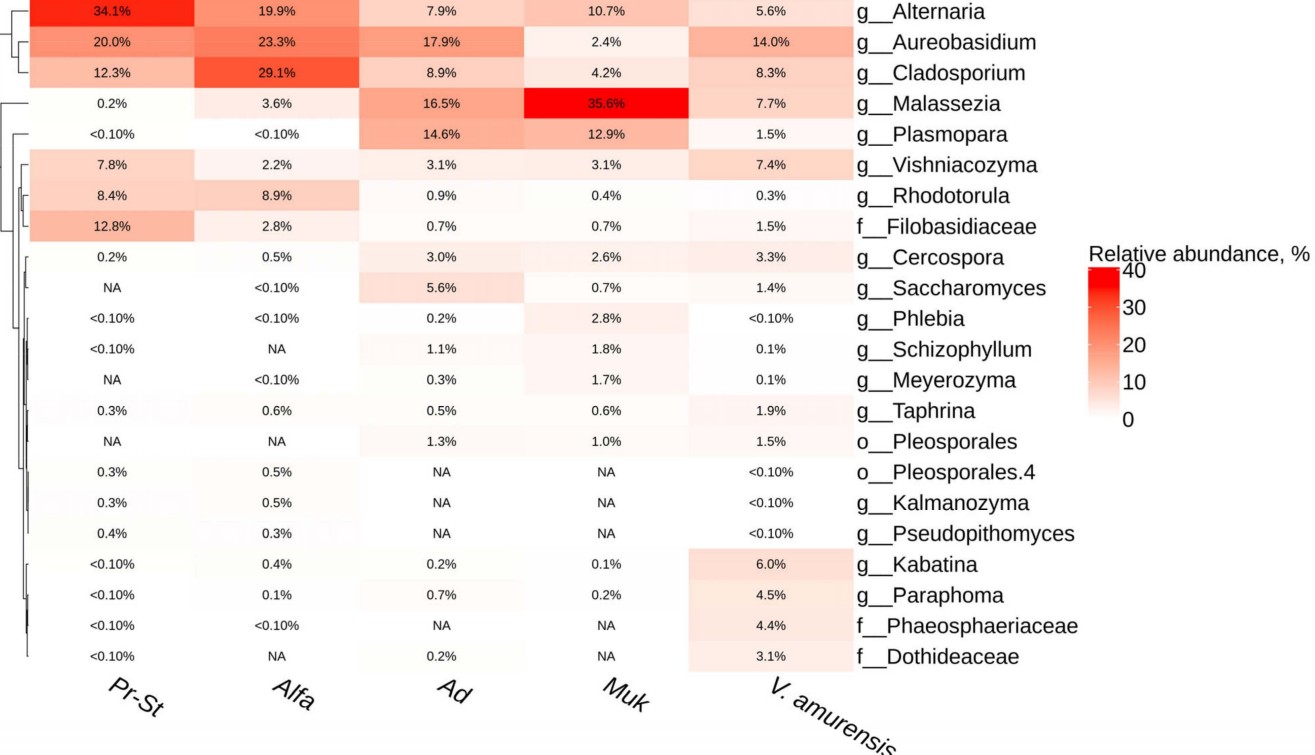

**Figure 4.** Genus-level relative endophytic fungi and fungi-like microorganism abundance heat maps of significant taxa according to next-generation sequencing (NGS) in grapevine cultivars and wild

grape *Vitis amurensis*. Pr-St—*Vitis* Elmer Swenson 2-7-13 cv. Prairie Star from commercial vineyard PRIM ORGANICA; Alfa—*Vitis labrusca* × *Vitis riparia* cv. Alfa from PRIM ORGANICA; Ad—*Vitis vinifera* × *V. amurensis* cv. Adele from commercial vineyard Makarevich; Muk—*V. riparia* × *V. vinifera* cv. Mukuzani; *V.amurensis*—the sum of data for all *V. amurensis* grapevines obtained early [39]. The top 10 most abundant taxa from each grapevine are displayed. White squares (NA) represent the absence of taxa.

### 3.4. A Comparative Analysis of the Endophytic Microbial Communities from Cultivated and Wild Grapevine Varietes

A comparative analysis of the endophytic communities of the cultivated grapevine varieties analyzed in this study with the previously studied endophytic biodiversity in wild grape *V. amurensis* growing in the Far East of Russia was carried out. A total of 7,462,656 *16S* and 845,338 *ITS1* paired reads were obtained from 50 samples of wild grape *V. amurensis* (Supplementary Materials Tables S1 and S2).

Figure 5a,b shows the results of the analysis of the alpha and beta bacterial endophytic diversity, respectively. The grapevine cultivar samples are not statistically different based on alpha diversity compared to the samples of the wild grape *V. amurensis* (Figure 5a).

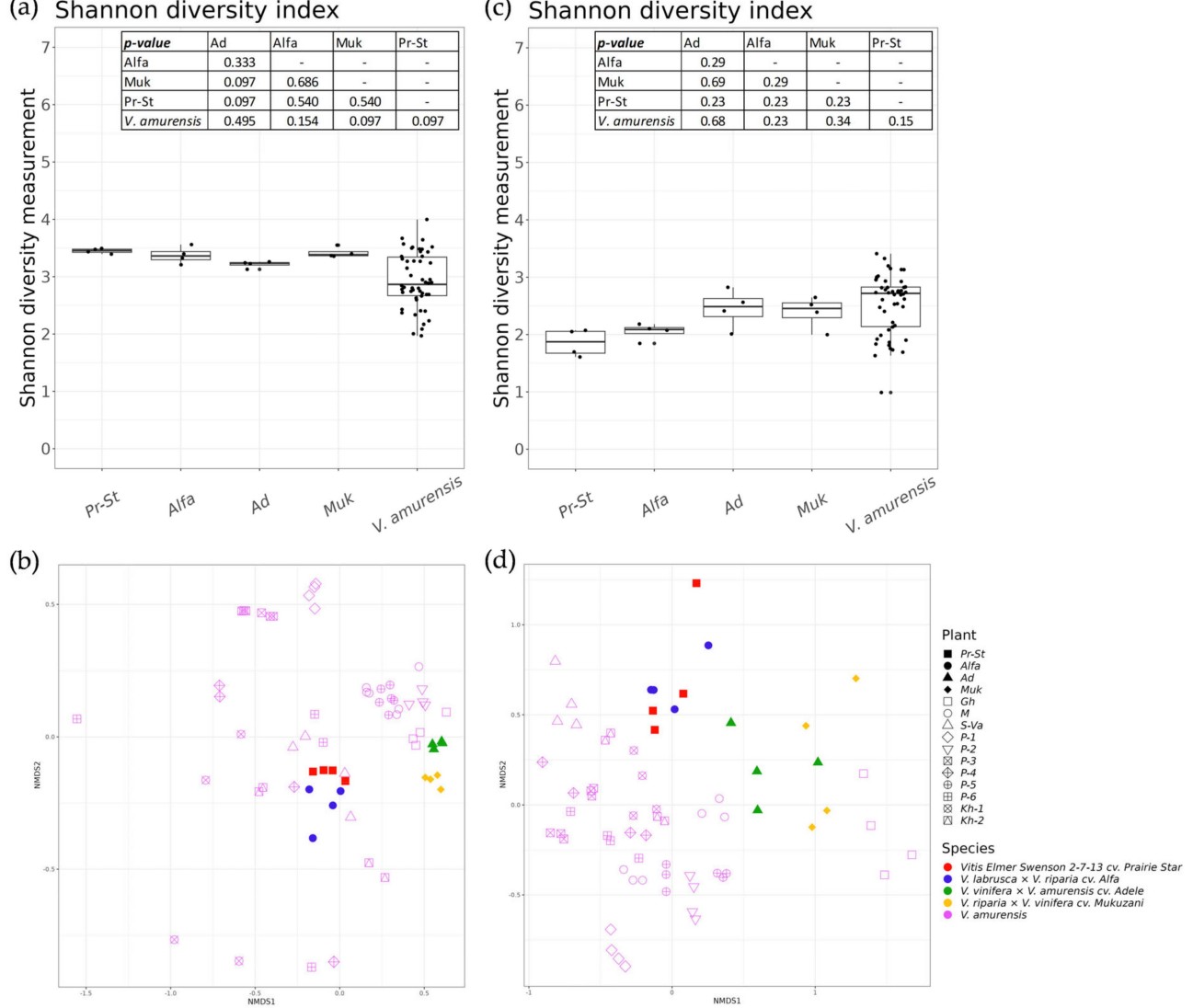

**Figure 5.** A comparison of endophytic bacterial, fungi, and fungi-like microorganism communities of grapevine cultivars and wild grape *Vitis amurensis*. Pr-St—*Vitis* Elmer Swenson 2-7-13 cv. Prairie Star from commercial vineyard PRIM ORGANICA; Alfa—*Vitis labrusca* × *Vitis riparia* cv. Alfa from

PRIM ORGANICA; Ad—*Vitis vinifera* × *V. amurensis* cv. Adele from commercial vineyard Makarevich; Muk—*V. riparia* × *V. vinifera* cv. Mukuzani; *V.amurensis*—the sum of data for all *V. amurensis* grapevines obtained early [39]: (**a**) Shannon's alpha diversity boxplot of bacterial communities; (**b**) Bray–Curtis beta diversity NMDS plot of bacterial communities; (**c**) Shannon's alpha diversity boxplot of fungi and fungi-like microorganism communities; (**d**) Bray–Curtis beta diversity NMDS plot of fungi and fungi-like microorganism communities.

The beta diversity was examined using non-metric multidimensional scaling (NMDS) ordination to compare the significant differences and understand the clustering of samples between groups (Figure 5b). The NMDS ordination showed that samples of the grape cultivars were located in separate small clusters, while the *V. amurensis* samples were more distributed. Samples from one vineyard are closer to each other according to the NMDS ordination plot (Figure 5b). The PERMANOVA test demonstrated that a significant proportion of the variance between bacterial communities is explained when comparing grape varieties with *V. amurensis* samples (20% of variance, $p < 0.001$). The factors "location of plants" and "individual plants" explain 50% and 10% of variance between the endophytic microbiomes, respectively ($p < 0.001$) (Figure 5b and Supplementary Materials Table S5).

Figure 5c,d shows the results of the alpha and beta diversity analysis of the fungi and fungal-like endophytic biodiversity of different grape samples. The alpha diversity of the grape variety samples was not statistically different among each other and from the wild grape *V. amurensis* samples. Also, the Shannon diversity index median value was slightly lower in the Prairie Star and Alfa grape varietes from PRIM ORGANICA than the median value in *V. amurensis*, whereas the medians were similar in the *V. amurensis* samples and the grapes Adel and Mukuzani from the Makarevich vineyard (Figure 5c). NMDS ordination showed that samples of grape cultivars from one vineyard are closer to each other (Figure 5d). The PERMANOVA test demonstrated that the grape cultivars and *V. amurensis* samples were significantly different based on beta diversity (18% of variance, $p < 0.001$). The "location of plants" and "individual plants" factors explain 56% and 3% of the variance between endophytic mycobiomes, respectively ($p < 0.001$) (Figure 5d and Supplementary Materials Table S6).

### 3.5. Analysis of Associations of Endophytic Bacteria and Fungi in the Vitis Microbiome

Using the microbiome network method, we analyzed the effect of endophytic communities of bacteria and fungi on each other in visually healthy *Vitis* plants, based on the data presented in Supplementary Materials Tables S1 and S2.

According to the analysis of the microbiome networks, the largest number of associations are observed between bacteria (196) (Figure 6a,b). The number of positive associations is ~three times higher than the number of negative ones (154 "+" vs. 42 "−"). Moreover, the number of positive and negative edges between bacteria and fungi is approximately the same (11 "+" and 12 "−") and amounts to ~12% of the total number of associations. On the other hand, fungi are significantly less connected to each other compared to bacteria, and have approximately the same number of positive and negative associations with each other (8 "+" and 6 "−") (Figure 6b).

Among bacteria, three classes are distinguished in descending order according to the number of positive and negative connections between the taxa at the genus level, Gammaproteobacteria, Actinobacteria, and Alphaproteobacteria, and among fungi—the class Dothideomycetes (Figure 6c). Gamaproteobacteria and Actinobacteria have the highest number of positive intergeneric relationships within their class. The genus-level taxa of the class Actinobacteria have the largest number of negative relationships compared to other classes.

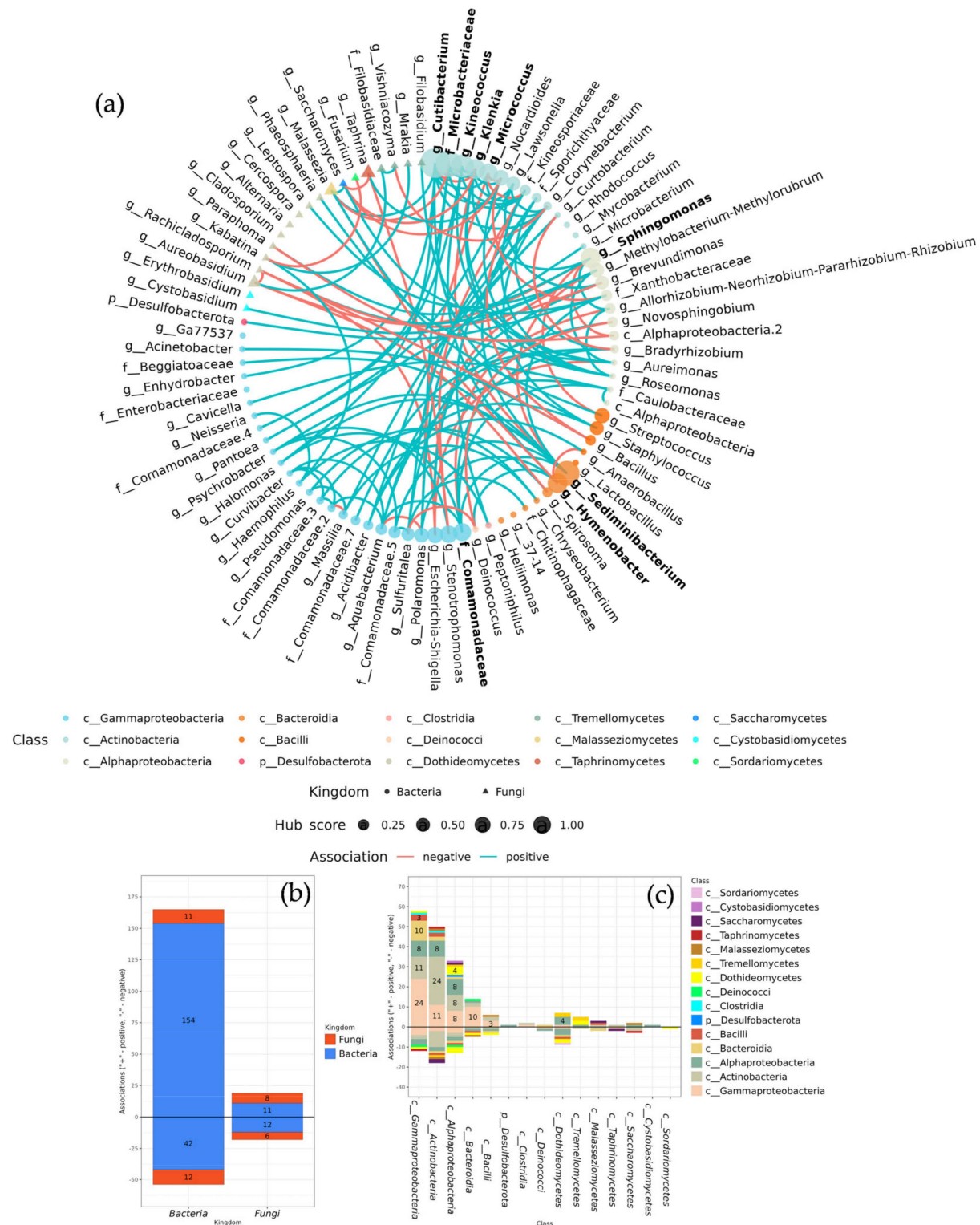

**Figure 6.** Positive and negative associations of endophytic bacteria and fungi in the *Vitis* microbiome. (**a**) Interdomain ecological networks of the bacterial–fungal associations; (**b**) summary bar plot of kingdom-level bacteria-fungi associations; (**c**) summary bar plot of class-level bacteria–fungi associations. The top 10% taxa of genus level based on the values of Kleinberg's hub centrality score were qualified as hubs. In (**a**), hub taxa are shown in bold.

We then discovered hub taxa in the *Vitis* endophytic microbial community, which have an important influence on the connectivity between the bacterial and fungal taxa

(Figure 6a). The class Actinobacteria is characterized by the largest number of genus-level taxa that we marked as hubs, viz *Cutibacterium*, *Microbacteriaceae*, *Kineococcus*, *Klenkia*, and *Micrococcus*. The Bacteroidia class has two hubs: *Sediminibacterium* and *Hymenobacter*. Alphaproteobacteria and Gammaproteobacteria are characterized by *Sphingomonas* and *Comamonadaceae* hubs, respectively. No hubs were found in the fungi.

## 4. Discussion

The environmental conditions in any wine-growing area can affect the quality of the grapes and lead to variations in the taste and aroma of the wine, even for the same grape varieties [40]. New research has suggested that specific microbial communities associated with *V. vinifera* may be a key element of the terroir since microbial processes essential to grape growing and wine production show spatial patterns linked to the vineyard site [41]. Also, to reduce plant pathogens and the negative impact of agricultural practices on the environment, biological control is an increasingly successful and widespread strategy. Beneficial microbes, especially endophytic bacteria and fungi, are used to counteract plant pathogens and limit the use of agrochemicals [42].

This study aimed at characterizing the endophytic microbiome of the most common grape varieties of Primorsky Krai, Russia. In general, the endophytic bacterial community composition was presented by the classes Gammaproteobacteria, Alphaproteobacteria, Bacteroidia, and Actinobacteria. The general biodiversity profile of endophytic bacteria correlated with the data from previous studies [20,40]. The most dominant taxon for the grapevines cvs. Adel and Mukuzani from Makarevich was the taxon Comamonadaceae (15–26%) (Figure 2). It is known that the family *Comamonadaceae*, which are the dominant denitrifiers, might play a primary role in improving the denitrification performance of the environment [43]. Perhaps an increase in the percentage of this taxon of endophytic bacteria indicates an excessive amount of nitrates in the soil of the Makarevich vineyard. Also, in the samples of the grapevine cvs. Adel and Mukuzani, endophytic bacteria *Aquabacterium* (9–14%) and *Spirosoma* (7–11%) were found in large numbers (Figure 2). It has been shown early that *Aquabacterium* sp. A7Y can produce endo-chitosanase *AqCoA*, which degrades the fungal cell walls, and the resulting oligosaccharides are promising weapons to protect plants from fungal disease [44]. Also, it was shown that *Spirosoma* sp. are radiation-resistant [45,46] and propanil-degrading bacteria [47]. The community of endophytic bacteria was represented in a significant percentage by the genera *Sphingomonas* and *Methylobacterium-Methylorubrum* in the samples of grape cvs. Prairie Star and Alpha from the vineyard PRIM ORGANICA and in the wild grape *V. amurensis* (Figure 2). The presence of *Sphingomonas* and *Methylobacterium-Methylorubrum* bacteria in wines can affect their sensory qualities due to the production of volatile organic compounds, and these compounds contribute to the unique characteristics of the wines [48].

The dominant classes of fungi were Dothideomycetes and Malasseziomycetes in all analyzed samples, which correlated with the endophytic community of fungi in grapes from a vineyard in China [49]. In samples of grapes from the Makarevich vineyard, we identified characteristic dominant genera typically linked with wine and grapes (*Malassezia*) [49] along with several pathogenic fungi-like genera (*Plasmopara*) (Figure 4). It is known that powdery mildew, caused by *Plasmopara viticola*, is a severe disease that leads to significant grape harvest losses worldwide. However, certain American and Asian *Vitis* species exhibit varying levels of resistance to *P. viticola*. Some species like *V. rupestris* show moderate resistance, while others like *V. rubra*, *V. candicans*, *V. amurensis*, *V. riparia*, *V. cinerea*, or *Muscadinia rotundifolia* display high resistance [50–52]. Surprisingly, *P. viticola* can exist within the internal tissues of certain grape species or varieties without causing visible symptoms of powdery mildew. To control fungal-like organisms such as oomycetes (to which *P. viticola* belongs), the regular application of fungicides is necessary to prevent damage and economic losses [53]. However, the excessive use of fungicides can lead to the development of *P. viticola* strains that are resistant to these chemicals [54,55]. In the Makarevich vineyard, the presence of *Plasmopara* in the grape samples could indicate

that either the analyzed grape varieties are resistant to this oomycete or that the presence of *P. viticola* in the tissues does not necessarily result in the manifestation of powdery mildew symptoms. It is likely that continuous chemical treatments against oomycetes at the Makarevich vineyard contribute to a reduction in the number of *P. viticola* within the grape tissues, preventing the appearance of symptoms. However, these treatments may not completely eliminate the pathogen, allowing it to persist within grape tissues as an endophyte. The near absence of *Plasmopara* representation in the samples collected at the PRIM ORGANICA vineyard indicates a more effective treatment against this oomycete.

Also, for the grapevines from PRIM ORGANICA, the predominant genera were *Alternaria*, *Aureobasidium*, and *Cladosporium*. Notably, *Alternaria* is considered one of the main mycobiota populations of grapes at harvest [56,57]. Additionally, the pathogenic fungus *Alternaria alternata* is known to cause significant post-harvest losses in grapes. This issue has been extensively documented in various reports [58]. Perhaps the analyzed grape samples contained other *Alternaria* species that occupied the same ecological niches without causing crop losses. Also, contrasting findings indicate that several species of *Alternaria* can control the growth of different pathogens such as *Rhizoctonia solani*, *Fusarium oxysporum*, *B. cinerea*, and *Pseudomonas aeruginosa* [59,60]. As in our study, *Aureobasidium* was the predominant genus also in grapes in other studies [61,62]. It was shown that *Aureobasidium pullulans* had enzyme activity, such as pectinases, xylanases and cellulases, which encourages future studies regarding their application in winemaking, in particular, in improving the color extraction, technological parameters, and antioxidant activity of wine [63]. In addition, it is known that the *Cladosporium* sp. is indirectly involved in food spoilage because it produces mycotoxins [64], and their effect on wine quality is due to grape damage. Also, the fungus *Cladosporium* can withstand a high sugar content and low moisture [49]. Thus, the presence of dominant endophytic fungi of the genera *Alternaria*, *Aureobasidium*, and *Cladosporium* is an indicator that the selected grape varieties are grown in conditions of high humidity, and also that they can be used for winemaking. Also, samples of varieties from the Makarevich vineyard contained unique fungi genera such as *Resinicium*, *Ustilago*, *Ascotricha* (cv. Adele), *Daedalea*, and *Nothophoma* (cv. Mukuzani) (Supplementary Materials Table S3). It is known that fungi of the genus *Resinicium* is a worldwide genus of corticoid wood-inhabiting fungi [65], and *Ustilago* is the causative agent of corn smut disease and the culprit of considerable losses in grain yields [66]. Also, a unique taxon, *Bipolaris*, was found in the samples of the cultivar Alpha from PRIM ORGANICA (Supplementary Materials Table S3). *Bipolaris sorokiniana* (teleomorph, *Cochliobolus sativus*) is a wheat pathogen [67]. Perhaps the presence of these taxa is due to the proximity of cereal agricultural crops. Purnomo et al. [68] reported that the fungus *Daedalea dickinsii* has the capability to degrade dichlorodiphenyltrichloroethane (DDT, organochlorine pesticides) via a Fenton reaction. The presence of this taxon in the samples collected at the Makarevich vineyard may be indirect evidence of the presence of DDT in the soil, but this assumption needs to be checked.

According to the comparative analysis of cultivated grape varieties and wild grape *V. amurensis* endophytic communities, it was found that 30 and 87 unique genera of endophytic bacteria and fungi are represented in wild grapes, respectively (Figures 1b and 3b and Supplementary Materials Tables S3 and S4). The microbial community found in grapes can vary greatly, mainly due to external factors like the environment, location, and specific characteristics of grape varieties [69]. Previous studies have demonstrated that the types and quantities of endophytic bacteria and fungi living inside wild *V. amurensis* grapes in the Far East of Russia differ significantly depending on the plant's organs and weather conditions [20,21]. Leaves and stems tend to have the highest number of these microorganisms. The presence of an average temperature around 15 °C and ample precipitation seem to contribute to both the diversity and abundance of endophytic bacteria and fungi in *V. amurensis*. Conversely, hot and dry weather can lead to a significant decrease in the number of these microorganisms. Additionally, the use of various chemical treatments can also have a significant impact on the composition of the grape's endophytic community.

The lowest representation of taxa of endophytic bacteria and fungi compared with the wild grape *V. amurensis* is explained by the annual treatment of vineyards with chemicals against grape pathogens. However, in addition to pathogenic microorganisms, the wild grape *V. amurensis* may contain endophytic microorganisms that help the vine to withstand abiotic and biotic stresses. Therefore, the study of endophytes unique to wild grapes is an interesting task that requires future research.

Understanding the structure and nature of associations in the microbiome of healthy grapevines is important for the development of methods to make both wild and cultivated forms of *Vitis* more resistant to biotic stresses. The analysis of microbial associations between bacteria and fungi in visually healthy grapevine plant communities revealed mainly positive associations between bacteria–bacteria interactions and only a relatively small number of bacteria–fungi and fungi–fungi interactions. The classes Gammaproteobacteria, Actinobacteria, and Alphaproteobacteria had the strongest influence on the connectivity of the microbiome network. In microbial communities, hub taxa, characterized by high hub scores, are important as their removal can significantly affect the connectivity of the network [70,71]. Notably, the Actinobacteria class has the highest number of hub-taxa in our data, namely *Cutibacterium*, *Microbacteriaceae*, *Kineococcus*, *Klenkia*, and *Micrococcus*. Actinobacteria are known to be actively involved in stimulating plant growth and increasing disease resistance via beneficial interactions with the plant organism [72]. Further research is needed to explore the intriguing study of endophytic bacterial hubs of healthy grapes in the context of their role in microbiome networks associated with *Vitis* diseases.

## 5. Conclusions

This research focused on examining the diversity of endophytic bacteria and fungi found in cultivated grape varieties grown in the vineyards of Primorsky Krai, Russia. It was the first study of its kind in this region. According to our findings, around 18–20% of the variation in endophytic communities can be attributed to the disparities between cultivated and wild grapevines. Additionally, plant location and individual plants contribute to 50–56% and 3–10% of the variation, respectively. This suggests that factors like the environment in which the plants are grown and the specific characteristics of each plant play a significant role in shaping the diversity of the endophytic communities in grapevines. In addition, a comprehensive analysis was undertaken to investigate the microbial composition and inherent relationships within the grapevine microbiome. This research has significant potential to advance the use of endophytic microorganisms to enhance grapevine productivity and improve the resilience of cultivated *Vitis* species to various biotic and abiotic stresses.

**Supplementary Materials:** The following supporting information can be downloaded at https://www.mdpi.com/article/10.3390/horticulturae9121257/s1: Table S1: *16S* data samples used in analysis; Table S2: *ITS* data samples used in analysis; Table S3: Intersections in *16S* metagenome data of cultivars and *V. amurensis* plant samples; Table S4: Intersections in *ITS* metagenome data of cultivars and *V. amurensis* plant samples; Table S5: PERMANOVA results (*16s* data); Table S6: PERMANOVA results (*ITS* data).

**Author Contributions:** O.A.A. and K.V.K. performed the research design, data analysis, paper preparation, and experimental process. O.A.A., A.R.S. and P.A.C. collected the material. O.A.A., A.A.A., A.R.S., A.A.D., A.A.B. and N.N.N. performed the isolation of DNA for NGS. N.N.N. and Z.V.O. performed the bioinformatic analysis and visualization. A.S.D. was responsible for the writing—review and editing. All authors have read and agreed to the published version of the manuscript.

**Funding:** This work was supported by a grant from the Russian Science Foundation (grant number 22–74–10001, https://rscf.ru/project/22-74-10001, accessed on 18 October 2023).

**Institutional Review Board Statement:** Not applicable.

**Informed Consent Statement:** Not applicable.

**Data Availability Statement:** The data presented in this study are available within the article and Supplementary Materials.

**Acknowledgments:** We would like to thank Sergei Makarevich and Alexandr and Anna Storozhenko for their primary contribution to the sample collection for the present study.

**Conflicts of Interest:** The authors declare no conflict of interest.

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
