# Peer review of "Bacterial and Fungal Endophytes of Grapevine Cultivars Growing in Primorsky Krai of Russia"

_horticulturae, doi:10.3390/horticulturae9121257_

Round 1

Reviewer 1 Report

Comments and Suggestions for Authors

I suggest that another consideration is needed after the major revision. There are many inaccuracies in the scientific content of the manuscript. Incomplete data analysis, Introduction, Discussion and Conclusions require new insights based on the manuscript's results.

Details are given to the authors in the review.

Author Response

November 14th, 2023

Dear Editor,

Please find enclosed the revised manuscript (Ms) “Bacterial and fungal endophytes of grapevine cultivars growing in Primorsky krai of Russia” (horticulturae-2698084). We would like to thank the Reviewers for their time and comments on our work. We carefully examined the Ms according to the Reviewers’ remarks and addressed their comments in our revised Ms using track changes in Word with a brief point by point response to each comment provided separately. We hope that the revised version of the Ms will be suitable for publication in Horticulturae.

Thank you very much for considering our article and we are looking forward to your response.

Sincerely,

Olga A. Aleynova.

Review Report Form Reviewer #1

  1. The Introduction section provides a broad review of the literature on the use of individual bacterial species. However, this article analyzes only the bacterial and fungal overall microbiota of Vitis plants at the genus level. There is nothing in the review about the causality of the microbiota - the peculiarities of the formation of colonies in different parts of the plant, the change in the composition of colonies in perennial cultivated and wild plants, the dependence of the microbiota on environmental factors, and others. A lot of attention was paid to the bacteria that influence the biotic stress response in the plant, while the abiotic one was reviewed briefly, the importance of the reviewed literature should be balanced.

- Answer: Thank you for your valuable comment. We have added to the review a part about the causal relationship of microbiota - the features of colony formation in different parts of grapes. Please see lines 42-52.

Unfortunately, there is little information that indicates the influence of endophytic bacteria of grapes on resistance to abiotic stresses. We have provided all the data that are known to present date. Therefore, we have removed from the review the part devoted to the participation of endophytic bacteria of grapes in resistance to biotic stress. We deleted:” the endophytic strain Bacillus K1 of wild grape had high antifungal activity against Botrytis cinerea (gray mold) both in vitro and in vivo [11].”

  1. In Section 2.1, line 99 would need an explanation of which plants you considered "asymptomatic" or change to "plants healthy and normally developed". The methodology needs to be supplemented: what agrotechnical measures were applied in the plantations and the greenhouse.

- Answer:  We have changed the term "asymptomatic" to " Healthy and normally developed grapevine tissues ". We also added information about agrotechnical measures that were used in the vineyards. Please see lines 428-449.

  1. In the Results section and in other parts of the manuscript, "16S" should be corrected to "16S rRNA", and more precisely "16S rRNA V4 region", since only this part of the gene was used to identify bacteria. It should be noted that short fragment of gene is variable, but only allows identification of bacteria at the genus level in many cases. Also, even according to the whole gene 16 rRNA species are not identified in all cases (Metagenomic shogun sequencing is applied when naming species).

- Answer: We clarified the name of the “16S rRNA V4 region (16S)” and “ITS1-ITS2 rDNA region (ITS1)” gene parts in the "Results" section and introduced their abbreviations for further mention in the text. Please see lines 629-630.

  1. Please mention the results of in vitro control of epiphytic microorganisms.

- Answer: As a control of epiphytic microorganisms in vitro we used a sample of the last wash water (100 µL) and incubated on R2A and potato dextrose agar (PDA) plates. The absence of colony growth served to control the absence of epiphytic microorganisms in the samples. We have added this part to the "materials and methods" section, please see the lines 474-480.

  1. In this manuscript, only summary results per genotype are presented. The results need to be supplemented - please perform additional data analyzes and show the microbiota differences depending on the growth location (field, geographical area, greenhouse) and plant part (leaves to stems) (according to the study objects indicated in Tables SI and S2).

- Answer: We have changed the barplots of endophytic bacteria and fungi depending on the plant part (leaf and stem) and added a description to the new figures. Please see Figures 1a and 2a. We have considered the differences according to geographical location in a previously published article (Aleynova, O.A.; Nityagovsky, N.N.; Ananev, A.A.; Suprun, A.R.; Ogneva, Z.V.; Dneprovskaya, A.A.; Beresh, A.A.; Tyunin, A.P.; Dubrovina, A.S.; Kiselev, K.V. The Endophytic Microbiome of Wild Grapevines Vitis amurensis Rupr. and Vitis coignetiae Pulliat Growing in the Russian Far East. Plants 2023, 12, 2952. https://doi.org/10.3390/plants12162952), where the influence of geographical location is described in detail.

  1. Since plant bacterial endophytes are known to have fungicidal properties, evaluate how the diversity of endophytic bacteria affected the diversity of the endophytic fungal colony in the Vitis

- Answer: We have added a new chapter «3.5. Analysis of associations of endophytic bacteria and fungi in the Vitis microbiome» to the results section. Here we studied the interaction of fungi and bacteria in grapes and discussed it in Discussion chapter, please see the lines 1452-1447.

  1. After completing the results, expand the Discussion section appropriately. The current version of the Discussion is general and does not reflect the presented results. A speculative interpretation of the data is delivered, based more on the authors' assumptions than the results. For example, the author's discussion on the functions of metabolomics needs to be revised, as the results show only metabiota characteristics at the genus level. Synthesis of endo-chitosanases is characteristic of many bacteria and fungi, not only Aquabacterium Radiation resistance is also not unique to Spirosoma sp. (https://doi.Org/10.1016/j.foodchem.2013.10.083; https://doi.Org/10.1016/j.jenvrad.2021.106696).

- Answer: We rewrote the discussion according to the more detailed results. You can see the changes in the discussion using track changes in Word. We also decided to leave the part dedicated to Aquabacterium sp. and Spirosoma sp. We know that many bacteria and fungi have similar properties, but these interesting properties of certain types of grape endophytes may be beneficial to the grapevine.

  1. The paragraph (lines 415-422) requires clarification. In Figures la and 2a, it is difficult to see the unique genus of microorganisms identified by amurensis. An additional table or appendix with a list is required or otherwise clearly demonstrates uniqueness.

- Answer: We apologize for the inaccuracy in the text. Here it was necessary to mention the UpSet diagrams in Figures lb and 2b, as well as about Supporting Information Table S3 and Supporting Information Table S4. We have corrected this inaccuracy in the text, please see the lines 1434-1435.

  1. Also, explain the statement in more detail how and why the colony of endophytic microorganisms in the plant tissues can change when the experiment is repeated the following year. The chemical protection of the grape plants used in the study is not explained in the methodology, so the discussion is speculation on this claim.

- Answer: Thank you for your remark. We have added to the discussion a part of what factors can affect the biodiversity of endophytes, in addition to chemical treatment. Also, we have added a part on chemical processing by the vineyard to the materials and methods. Therefore, we believe that the statement about the effect of chemical protection of grapes on the endophytic community of grapes is appropriate. Please see the lines 1435-1446.

  1. Conclusions - not relevant to this publication based on the results, should be rewritten. There is now a summary text of knowledge not based on the purpose and data of this manuscript. Explain what results show that the grape cultivars contain endophytic microorganisms that can be used for winemaking. Can you name them? Specify the number of genera of endophytic bacteria and fungi in wild grape amurensis plants, better in different tissues and geographic locations. The last statement of conclusions is speculative, not supported by results also.

- Answer: We have rewritten the conclusion according to the results obtained

Reviewer 2 Report

Comments and Suggestions for Authors

Dear Authors,

in my opinion your work is very interesting in a cognitive context and contributes a lot to current trends towards alternative methods of control, yield and grapes quality of fruits, metagenomics and evolutionary taxonomy.

All tables and figures are appropriate for this type of article. In general, the paper has a logical flow and it is refined in detail. The abstract well correspond with the main aspects of the work. Nevertheless, I see a few and non-significant weak points in this work (given below), which I am convinced that the Authors are able to resolve very fast.

Confusing is the part of methodology describing procedure of preparation tissue's samples to Ilumina. I don't understand why authors after dissinfection incubated fragments of tissues on media in Petri Dishes? In my opinion it is useless since first of all there was not indicated the period of this treatment, secondarily, sooner or later there were the groth of endophytic colonies around the tissues. I am looking forward for explanation this part of methodology. Besides I would be sceptical as regards to endophytic strans representing genus Plasmopara, since it is known that Plasmopara viticola is one of the most harmful pathogen of Vitis vinifera, and there is a thin line beetwen pathogenicity and endophytism, and authors should mention about this in the disscusion.

Moreover:

Line 425- should be "first"

Author Response

November 14th, 2023

Dear Editor,

Please find enclosed the revised manuscript (Ms) “Bacterial and fungal endophytes of grapevine cultivars growing in Primorsky krai of Russia” (horticulturae-2698084). We would like to thank the Reviewers for their time and comments on our work. We carefully examined the Ms according to the Reviewers’ remarks and addressed their comments in our revised Ms using track changes in Word with a brief point by point response to each comment provided separately. We hope that the revised version of the Ms will be suitable for publication in Horticulturae.

Thank you very much for considering our article and we are looking forward to your response.

Sincerely,

Olga A. Aleynova.

Review Report Form Reviewer #2

Dear Authors,

in my opinion your work is very interesting in a cognitive context and contributes a lot to current trends towards alternative methods of control, yield and grapes quality of fruits, metagenomics and evolutionary taxonomy.

All tables and figures are appropriate for this type of article. In general, the paper has a logical flow and it is refined in detail. The abstract well correspond with the main aspects of the work. Nevertheless, I see a few and non-significant weak points in this work (given below), which I am convinced that the Authors are able to resolve very fast.

-Answer: Dear Reviewer, we thank you for your careful consideration of our manuscript and high appreciation of the research done.

Confusing is the part of methodology describing procedure of preparation tissue's samples to Ilumina. I don't understand why authors after dissinfection incubated fragments of tissues on media in Petri Dishes? In my opinion it is useless since first of all there was not indicated the period of this treatment, secondarily, sooner or later there were the groth of endophytic colonies around the tissues. I am looking forward for explanation this part of methodology.

-Answer: To prepare grape samples for further NGS, we performed surface sterilization of grape tissues. To verify the success of this sterilization method, a sample of the last wash water (100 µL) was incubated on R2A and potato dextrose agar (PDA) plates. The purpose was to ensure that there was no colony growth on these plates, indicating the absence of any bacteria or fungi contamination from the outside. This sterilization procedure is crucial for maintaining the integrity and purity of the grapevine tissues, preventing any unwanted microorganisms (epiphytes) from interfering with subsequent analysis or experiments. We have added this part of the explanation to the new version of the manuscript. Please see the lines 474-480.

-Besides I would be sceptical as regards to endophytic strans representing genus Plasmopara, since it is known that Plasmopara viticola is one of the most harmful pathogen of Vitis vinifera, and there is a thin line beetwen pathogenicity and endophytism, and authors should mention about this in the disscusion.

-Answer: Thank you for your valuable comment. We have added a part dedicated to the pathogenicity and endophytism of Plasmopara to the discussion. Please see the lines 1360-1379.

Moreover:

Line 425- should be "first"

-Answer: Thank you for your remarks, we made this proposal the first in our manuscript. Please see abstract, lines 12-13

Round 2

Reviewer 1 Report

Comments and Suggestions for Authors

This version is suitable for publication, in my opinion.